ecology

age at first birth, age at maturity, Allee effect, *Canis lupus*, primiparity, inbreeding

**Author for correspondence:**
Camilla Wikenros
e-mail: camilla.wikenros@slu.se

†Present address: Department of Ecological Dynamics, Leibniz-Institute for Zoo and Wildlife Research, Berlin, Germany.

# Age at first reproduction in wolves: different patterns of density dependence for females and males

Camilla Wikenros[1], Morgane Gicquel[1,†], Barbara Zimmermann[2], Øystein Flagstad[3] and Mikael Åkesson[1]

[1]Grimsö Wildlife Research Station, Department of Ecology, Swedish University of Agricultural Sciences, 73993 Riddarhyttan, Sweden
[2]Faculty of Applied Ecology, Agricultural Sciences and Biotechnology, Campus Evenstad, Inland Norway University of Applied Sciences, 2480 Koppang, Norway
[3]Norwegian Institute for Nature Research, PO Box 5685 Torgard, 7485 Trondheim, Norway

CW, 0000-0002-2825-8834; MG, 0000-0002-0823-1239; BZ, 0000-0001-5133-9379; ØF, 0000-0002-5534-8069; MÅ, 0000-0002-4325-8840

Age at first reproduction constitutes a key life-history trait in animals and is evolutionarily shaped by fitness benefits and costs of delayed versus early reproduction. The understanding of how intrinsic and extrinsic changes affects age at first reproduction is crucial for conservation and management of threatened species because of its demographic effects on population growth and generation time. For a period of 40 years in the Scandinavian wolf (*Canis lupus*) population, including the recolonization phase, we estimated age at first successful reproduction (pup survival to at least three weeks of age) and examined how the variation among individuals was explained by sex, population size (from 1 to 74 packs), primiparous or multiparous origin, reproductive experience of the partner and inbreeding. Median age at first reproduction was 3 years for females ($n = 60$) and 2 years for males ($n = 74$), and ranged between 1 and 8–10 years of age ($n = 297$). Female age at first reproduction decreased with increasing population size, and increased with higher levels of inbreeding. The probability for males to reproduce later first decreased, reaching its minimum when the number of territories approached 40–60, and then increased with increasing population size. Inbreeding for males and reproductive experience of parents and partners for both sexes had overall weak effects on age at first reproduction. These results allow for more accurate parameter estimates when modelling population dynamics for management and conservation of small and vulnerable wolf populations, and show how humans through legal harvest and illegal hunting influence an important life-history trait like age at first reproduction.

## 1. Introduction

Age at first reproduction constitutes a key life-history trait in animals, and is evolutionarily shaped by fitness benefits and costs of delayed versus early maturation [1,2]. Knowledge about age at first reproduction and how it varies within species is crucial for conservation and management of threatened species because of its demographic effects on generation time and population growth [3–6]. Early maturity has the benefit of increasing reproductive output at young age, and selection is expected to favour individuals breeding early [1]. Still, apart from being constrained by the physiological capacity connected to reproduction and limited access to vacant mates and space, reproduction may also be delayed to save resources for further growth and future reproduction or to get access to a territory of higher quality [7–9]. At the individual level, factors such as population density, prey density and availability,

habitat quality, and varying risk of mortality in the landscape may thus all affect the maturation process [9–13].

Density-dependent age at first reproduction, with younger first-time breeders when population size is small, may have important consequences for the persistence of long-lived species as a buffer against population fluctuations [10]. For recently re-established or small populations, age at first reproduction is expected to be nonlinearly related to population density [14], similar to other fitness components. Initially, starting at the founder event, the population may be subject to an Allee effect [15]. The Allee effect predicts a decrease in age at first reproduction with population growth, as individual fitness may be facilitated by the presence of more conspecifics. When the population approaches carrying capacity and resources become limited for survival and/or reproduction, fitness decreases, with age at first reproduction being positively related to population density. Thus, the success of recolonization or reintroduction of species with initially low densities, possibly experiencing Allee effects, depends partly on age at first reproduction [16].

Since age at first reproduction often is intimately related to body weight [9,17], it is likely that the environment experienced as a juvenile, including parental provisioning, competition with siblings and social organization, influences when an individual has its reproductive debut. Offspring from a primiparous litter (i.e. first litter of the parents) compared with multiparous litters may be negatively affected, due to less experienced and/or incompletely developed parents [18]. In group living species, older siblings may also act as helpers and increase the fitness and speed up the physical development of younger siblings to attain sexual maturity by providing extra food, protection and training [19]. However, in canids the effect of older siblings on juvenile fitness traits shows inconsistent patterns and depends on the prevailing ecological conditions and its influence on food competition within family groups [20,21]. The presence of non-reproducing subadults and adults in the family group may thus speed up the maturation of juveniles during times of food surplus, while in turn slow down development when resources are limited [22].

The wolf (*Canis lupus*) is a territorial and group living species that has recolonized parts of Europe [23] and North America [24] during the last decades. Still, the ranges of the wolf populations are highly fragmented and sometimes populations are small and inbred, making them vulnerable to stochastic demographic effects [25–27]. This puts emphasis on the species' ability to disperse and reproduce. Even though the social ecology of wolves has been extensively studied for almost half a century [28], there is relatively little information on age at first reproduction and the factors that affect this trait. On rare occasions, breeding wolves may be as young as ten months old [29,30], but most wild wolves seem to start reproducing at two years of age [31–34], whereas in some areas female wolves do not normally breed until four years of age [35,36]. Mech *et al*. [37] showed that two wolf populations differing in age at first reproduction also differed in age of maximum body mass, which could be due to genetic differences or differences in food availability. The latter is documented [9,11] but effects of genetics on age at first reproduction remain to be tested.

The wolf population on the Scandinavian Peninsula is suitable for the study of individual variation in age at first reproduction and the impact of different ecological factors that may affect this important trait for at least two reasons. First, it has been monitored since it was founded in the early 1980s, consisting of only one breeding territory during the first 8 years [38–40]. This was followed by a positive population growth until 2016/2017 when the population consisted of 74 territories with greater than or equal to 2 wolves [41], and a subsequent decrease to 72 territories in 2017/2018 [42]. This long-term study population is characterized by changing population size and breeding range, which opens up for potential changes in competition for space, resources and mates. Pair dissolution rates among the wolves have been high with an average probability of 0.32 for a pair to dissolve from one year to the next [43]. Also, after loss of one pair member, the lost partner was most often replaced by a new partner the following winter [43], and the replacement may be more likely to be a first-time breeder [12], but this differs between sexes [44]. Secondly, the population has been subject to fluctuating inbreeding levels, which gives us the opportunity to study the effect of inbreeding and immigration on age at first reproduction [45]. Inbreeding depression has been documented on several life-history traits linked to reproduction and possibly survival [40,45,46]. During the study period, an immigration event led to genetic rescue, with increased population growth as a result [45]. The probability of finding a partner and reproducing for immigrant offspring was more than twice that of inbred offspring of resident wolves [45]. Still, we did not investigate if immigrant offspring also mated earlier than resident wolves. Even though there are few studies on the effect of inbreeding on the rate of maturation in mammals, there are studies showing negative effects of inbreeding on growth rate [47,48] and body condition [49], where less developed individuals in poor condition may be expected to stay longer with their parents and thus delay their reproduction.

With data from the recently recolonized but rather isolated wolf population on the Scandinavian Peninsula, we gain better understanding of factors influencing age at first reproduction and how this changed during its re-establishment. We estimated the age at first successful reproduction with pup survival to at least three weeks of age and examined how the variation among wolves was affected by population size, primiparous or multiparous origin, reproductive experience of the partner, and inbreeding coefficient (F) as well as classification of individuals as first-, second- or third-generation immigrant descendants or resident origin. We have four predictions in relation to age at first reproduction. (1) Age at first reproduction is quadratically related to population size. At smaller population sizes, the ages are likely to be higher due to a predicted Allee effect during the population's recolonization phase and at larger populations size the ages are predicted to increase due to increasing competition for food and space. (2) Age at first reproduction is lower for wolves with multiparous than primiparous origin due to the benefits of parental learning and the potential access to more food from helpers and/or more experienced parents. (3) Age at first reproduction is lower for those having a partner with previous breeding experience assuming that individuals prefer to pair with an established mate rather than settling in a new unoccupied territory with an unexperienced mate. (4) Given that inbreeding may have a negative effect on individual body condition it is predicted to delay the age at first reproduction, especially when wolves are competing for space and mating opportunities. Knowledge on age at first

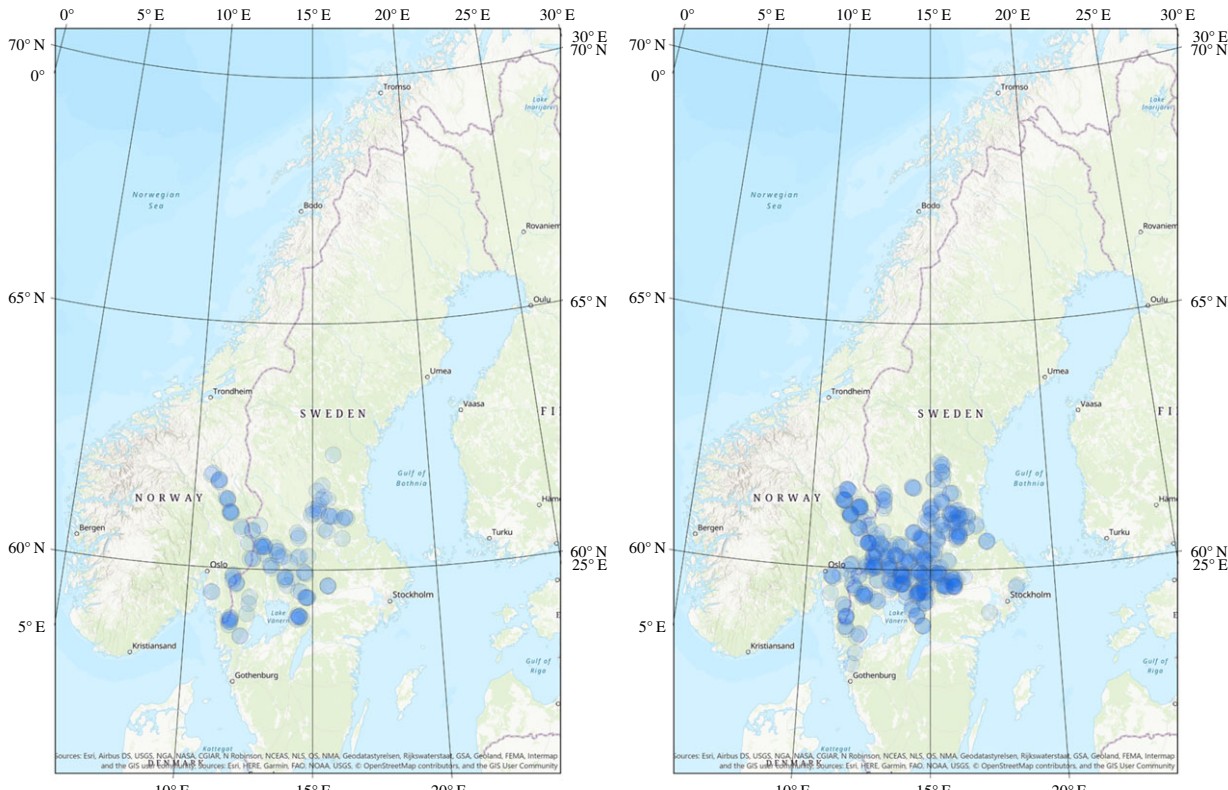

**Figure 1.** Breeding range of the Scandinavian wolf population during the winters of 1998/1999–2007/2008 (to the left) and 2008/2009–2017/2018 (to the right). Blue, transparent circles show confirmed reproductions with a 18 km buffer from the centroid location representing average territory size [50]. (Online version in colour.)

reproduction and factors affecting it will increase accuracy when modelling wolf population dynamics for small populations in need of conservation and management efforts.

## 2. Methods

### (a) Study system

The study was conducted on the Scandinavian Peninsula (south-eastern Norway and south-central Sweden; hereafter Scandinavia) in the boreal forest zone. The wolf was declared functionally extinct in Scandinavia in 1966. In 1983, two wolves from the Finnish–Russian population reproduced in a cross-border territory of Sweden and Norway, and thereby founded the current Scandinavian population [38,40]. The founding pair bred for 3 years and from 1987 to 1990 the population was maintained by incestuous reproductions resulting in inbreeding [40]. In 1991, wolves reproduced for the first time in two territories and the male in the second territory was a third Finnish–Russian immigrant [39]. By then, the wolf population started to increase in numbers and expand their breeding range in Scandinavia (figure 1). Later, five more immigrants successfully reproduced: two males in 2008, one translocated pair in 2013, one male in 2016 and one female in 2017 [39,40,45,51]. Licensed harvest of wolves occurred during January–February in 2010, 2011, 2015–2018 in Sweden, during 2005, 2007–2009, 2011–2018 in Norway, and protective culling occurred annually in both countries. Illegal hunting was the single most common mortality cause of Scandinavian wolves during 2000–2017 [52]. See Wikenros *et al.* [53] for further description of the study system.

### (b) Annual reproductions and population size

We used annual monitoring data of confirmed reproductions during 1978–2018. The aim of the joint cross-border wolf monitoring in Scandinavia has been to identify the annual number of all solitary resident wolves (until 2013, thereafter not obligatory), wolf pairs that scent-mark a territory, family groups (i.e. a group with at least three wolves) and annual reproductions [38,54]. We used the number of territories (i.e. the number of scent-marking pairs and family groups) during the monitoring season that preceded reproduction [38,42] as a proxy for population size.

### (c) Genetic analysis

Capture, handling and VHF/GPS collaring of wolves [55,56] was performed by the Scandinavian wolf research project SKANDULV and occasionally by management authorities in Norway and Sweden. Samples were derived both from dead and live-captured wolves, and from non-invasive samples collected during the annual Norwegian–Swedish wolf monitoring while snow-tracking [45]. Sex was determined either from morphological sexing of dead or captured individuals, or from DNA-analysis (see electronic supplementary material for further details).

To determine parental identities and to reconstruct the pedigree, we used a two-step process based on microsatellite genotypes and field observations in accordance with Åkesson *et al.* [45]. Of 452 breeding individuals, we were able to determine the population of origin and the parental identities of Scandinavian born individuals to previous genotyped individuals in 408 cases (90%). In 43 cases, the parental genotypes could be reconstructed to such a degree that the grandparents could be identified. That leaves one individual (less than 1%) were the genealogy could not be reconstructed. Based on the reconstructed pedigree, we calculated the inbreeding coefficient (F) of the breeding individuals using CFC v. 1.0 [57] and classified their relationship to immigrants as $F_1$, $F_2$ or $F_3$ (first-, second- or third-generation immigrant descendants, respectively) or native inbred (i.e. no close relationship with immigrants) [45]. Some individuals were excluded from the analyses, including one individual with unknown F and 11 individuals that reproduced the first time

with a parent and thus may have been influenced by other factors than the ones included in the study. Age at first reproduction for those 11 individuals (sex and years of first reproduction in subscript) was 1 ($n_{\male 2018} = 1$), 1–2 ($n_{\female 2002} = 1$, $n_{\male 2006} = 1$), 2 ($n_{\male 2004, 2015} = 2$, $n_{\male 2017} = 1$), 2–3 ($n_{\female 2018} = 1$, $n_{\male 2013} = 1$), 4 ($n_{\male 1991} = 1$) and 2–5 ($n_{\female 2017} = 2$) years.

## (d) Age determination

We used three sources of information to determine the year, or a range of years, of first reproduction of individuals including: (1) year of birth was given to offspring observed within the first year that the parents reproduced and to offspring with parents that only reproduced a single year, (2) the individual was identified as a pup at the den, or (3) the individual was identified and aged as a juvenile when captured during its first winter. A handled wolf was aged to less than 1 year using several combined methods, including no or little visible tooth wear, puppy fur, the presence of a juvenile-specific growth zone on the front leg (tibia) which disappears before 1½ year old [32]. Age at first reproduction was unknown for nine individuals consisting of immigrants or individuals with unknown parents. Since the vast majority of individuals that bred for their first time were either 2 years or 3 years of age, we also investigated the determinants of first-time breeding using two discrete age classes, including 1–2 years of age (defined as earlier first reproduction) versus greater than or equal to 3 years of age (defined as later first reproduction).

## (e) Primiparous or multiparous origin and partner experience

Primiparous litters were defined as litters where both parents were first-time breeders and multiparous litters when at least one of the parents had bred before. Individuals with known age were assumed to have primiparity origin or multiparity origin, respectively, when the year of birth did or did not coincide with the first year of parental reproduction according to monitoring data. Individuals with unknown year of birth were not included in the modelling analyses. Reproductive experience of the partner was a binary variable describing for a given first-time breeder whether the partner had bred before. This was based on yearly monitoring data on reproduction.

## (f) Statistical analyses

We conducted statistical analyses in R version 3.5.0 [58] using general linear models (GLM). We first modelled age at first reproduction (1–7 years old) using a quasi-Poisson link due to underdispersion (ratio of deviance to degrees of freedom of the full model = 0.24 for females and 0.26 for males). We then modelled the reproductive age category (later or earlier first reproduction) with a binomial link. For all models, we used population size (continuous variable), the quadratic term of population size, primiparity origin (2-level category: yes or no (only included with estimated age as response variable)), partner experience (2-level category: yes or no) and F (continuous variable) or immigrant relationship (4-level category: $F_1$, $F_2$, $F_3$, native inbred), as explanatory variables (see electronic supplementary material, table S1 for further details). We also included the interaction between population size and immigrant relationship. F and immigrant relationship were never included in the same models. All models that contained the quadratic effect of population size also accounted for the linear effect, and we tested all combinations of models. After experiencing convergence problems with mixed models using combined data from females and males with pair identity as random factor, we analysed females and males separately in order to avoid pseudo-replication of pairs. For the

analysis of age at first reproduction, we compared candidate models using quasi-AIC (QAIC), and for the reproductive age category we used the sample-size corrected Akaike information criterion ($AIC_c$) as well as AIC weights ($w_i$) from the 'MuMIn' package [59] in R. Models with $\Delta AIC \leq 2$ were used to generate model-averaged parameter estimates [60]. We used AIC weights on model sets with $\Delta AIC \leq 2$ to generate relative variable importance (RVI) weights for each explanatory variable.

# 3. Results

Age at first reproduction was given for 134 individuals, whereas for 297 individuals age at first reproduction was estimated with an uncertainty of 2 ($n = 120$), 3 ($n = 99$), or 4–9 years ($n = 78$) (electronic supplementary material, figure S1). Among these 297 wolves, 115 could be classified as earlier or later reproducers. Of the 134 wolves with exact age, 1% reproduced at 1 year of age (two males), 59% at 2 years, 28% at 3 years and 12% at 4 years or older. Of the 249 individuals (including the ones with exact years) which could be classified into earlier and later reproducers, 52% reproduced at the age of 1 or 2 years, and 48% reproduced at 3 years or later. The oldest age at first reproduction was 8–10 years for two females during 2001. Median age at first reproduction was 3 (range 2–7) and 2 (range 1–7) years for females ($n = 60$) and males ($n = 74$), respectively.

## (a) Age at first reproduction as continuous response

In females, age at first reproduction decreased with increasing population size (figure 2a). The inbreeding coefficient F had a positive correlation with age at first reproduction. Primiparity origin and partner experience were also included in the top models, but the standard error around the estimate (i.e. the confidence interval) of the effect included zero indicating weak effects on age at first reproduction (table 1). The immigrant relationship variable was not retained in the top models (electronic supplementary material, table S2).

For males, the top models of continuous age at first reproduction included a negative correlation with primiparity origin and weak relationships with population size, partner experience and F (all confidence intervals of the estimates included zero), and the highest-ranking model was the intercept model (figure 2b and table 1; electronic supplementary material, table S2).

## (b) Later or earlier first reproduction

In females, the top models for the probability to reproduce later included population size as linear term and partner experience, but those relationships were weak (all confidence intervals of the estimates included zero), and the intercept model was the highest ranking model (table 1 and figure 3a; electronic supplementary material, table S2).

In males, the best models contained population size as a polynomial term, with a U-shaped relationship between the probability to reproduce later and population size (table 1 and figure 3b). The inbreeding coefficient was also contained in the top models, but the confidence interval of the estimate included zero, indicating a weak effect (table 1). The immigrant relationship variable was not retained in the top models (electronic supplementary material, table S2).

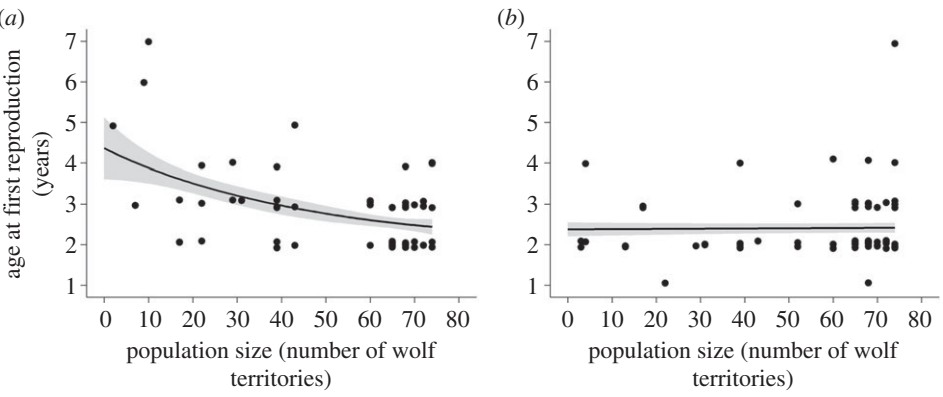

**Figure 2.** Age at first reproduction for (*a*) female and (*b*) male wolves in relation to population size (number of wolf territories) in Scandinavia (1992–2018; $n_{females} = 60$, $n_{males} = 74$). The lines indicate the fitted values, with associated standard errors, from the model-averaged estimates (table 1). Inbreeding coefficient was held at the mean, partner experience was held constant at 'no' and primiparity origin as 'yes'. Dots represent the observed values and for visual purposes overlapping dots are separated.

## 4. Discussion

There are few study populations of large mammals that are suitable for explaining the relative importance of intrinsic factors on important life-history traits, like age at first reproduction. With data spanning 40 years from a spatially isolated and newly founded population in Scandinavia we found that wolves reproduced at a relatively young age compared with other populations and that population size was the most important variable to explain the variation in age at first reproduction in both females and males. Contrary to our predictions we found only effects of the inbreeding coefficient for females, and overall weak effects of parent and partner experience.

In the Scandinavian wolf population, median age at first reproduction was 3 and 2 years of age for females and males, respectively, and 52–60% (depending on age estimate accuracy) of the individuals reproduced for the first time at the age of 1–2 years. Although rare, wolves may breed as yearlings [29,30]. In this study, three males reproduced as yearlings, two non-incestuously and one incestuously, all in the vicinity to or within their natal territory. In North American populations, wolves mostly start breeding at two years of age [31–34], but in some areas, female wolves do not normally breed until four years of age [35,36].

The treatment of age at first reproduction as either a continuous variable (years of age) or categorical variable (later versus earlier first reproduction) differed in its association with population size in females and males. Female age decreased with increasing population size, but the probability of later reproduction was only weakly related to population size. For males, the probability to reproduce later had a U-shaped relationship with population size. Generally, males showed a lower variation in age at first reproduction than females, with proportionally more males being either two or three years, indicating that a binomial model better explains the reproductive age of males than a Poisson model. By contrast, a Poisson model is likely better in explaining reproductive age for females than males since there was higher variation in age of later reproducers among females, variation that is lost when treating the data binomially.

In the early phase of the population's history, we found indications that the population was subject to an Allee effect in both females and males. The turning point between negative and positive density dependence of probability for later reproduction was as much as 40–60 territories. In Wisconsin and Michigan, USA, Stenglein & Van Deelen [61] estimated that a population crossed the Allee threshold at roughly 20 wolves in four to five packs. In our study, there was a turning point at a more than ten times higher wolf abundance, that coincided with an increasing turnover of territorial individuals [43,52]. Instead of being attributed to an Allee effect, where the low population density limits the access to partners, the long lasting decrease in age at first reproduction may therefore also be explained by an increase in turnover of territorial individuals possibly due to a higher incidence of illegal hunting [52]. Populations that experience high turnover of territorial individuals due to anthropogenic hunting are likely to experience a higher variation in age at first reproduction due to the varying access to resources for surviving non-territorial individuals [62]. Hunting may also have a negative impact on longevity of pair relations and increased access to unpaired territorial partners [43]. The disappearance rate of territorial pairs in the Scandinavian wolf population increased from 0.09 during the period 2000–2009 to 0.21 during the period 2010–2016, where the increased disappearance rate during the latter time period was mainly explained by illegal hunting [52]. Despite this high disappearance rate of territorial pairs, the number of territories increased from 65 to 74 during the last years of the study period (2012–2017). The high turnover of territorial pairs may be one of the reasons for the low age at first reproduction when territorial individuals are quickly replaced. Together with previous studies showing that the survival probability for non-territorial wolves is similar or lower than for paired individuals [52], we found indications that the replacement of lost mates and establishment of new territories is dependent on the number of young dispersers in the population and thus the reproductive output 2–3 years back. As the direct demographic contribution of immigrants is very low in this population, this can in turn have important implications for the population in situations if the reproductive output becomes reduced due to inbreeding and illegal hunting.

During the period when the probability for males to reproduce later was at its lowest (2007–2010, 39–60 wolf territories), two immigrant wolves (males) reproduced three years in a row, after a 17-year period without effective immigration. Average annual population growth before this event (2002–2007) was 13% and the corresponding number for the 2007–2012 period was 27% [45]. It is possible that this genetic rescue event increased the availability of mates, which partly

**Table 1.** GLMs to assess the effect of population size (PopSize) and the quadratic effect of population size (PopSize$^2$), primiparity origin (PrimOri), partner experience (PartExp) and inbreeding coefficient (F) for female and males on age at first reproduction of wolves in Scandinavia during 1988–2018. Analyses were conducted using age estimate (1–7) and age categories (later or earlier first reproduction) as the response variable. For each model, degrees of freedom (d.f.), difference in QAIC/AICc relative to the highest-ranked model ($\Delta$QAIC/AICc), and AIC weights ($w_i$) are shown. For simplicity, only models with $\Delta$QAIC/AICc $\leq 2$ (grey background), univariate models and intercept-only model are shown. Conditional model-averaged parameter estimates with standard error (s.e.) are shown for each variable retained in the best models ($\Delta$QAIC/AICc $\leq 2$). The reference in the analyses is 'yes' for primiparity origin, and 'no' for partner experience. Additionally, we used AICc weights on the full candidate model set to generate RVI for each explanatory variable.

| sex | dataset | intercept | PopSize | PopSize$^2$ | PrimOri | PartExp | F | d.f. | $\Delta$QAIC/AICc | $w_i$ |
|---|---|---|---|---|---|---|---|---|---|---|
| female | age (2–7) | | X | | | | | 2 | 0 | 0.18 |
| | $n = 60$ | | X | X | | | | 3 | 0.8 | 0.12 |
| | 24 models | | X | | | | X | 3 | 1.5 | 0.08 |
| | | | X | | X | | | 3 | 2.0 | 0.07 |
| | | | X | | | X | | 3 | 2.0 | 0.07 |
| | | X | | | | | | 1 | 2.4 | 0.05 |
| | | | | | | | X | 2 | 3.4 | 0.03 |
| | | | | | | X | | 2 | 4.4 | 0.02 |
| | | | | | X | | | 2 | 4.4 | 0.02 |
| | $\beta$ | 1.45 | −0.011 | 0.00022 | −0.032 | 0.014 | 0.56 | | | |
| | s.e. | 0.19 | 0.0092 | 0.00010 | 0.084 | 0.13 | 0.43 | | | |
| | RVI | | 0.84 | 0.34 | 0.27 | 0.27 | 0.34 | | | |
| | later/earlier | X | | | — | | | 1 | 0 | 0.16 |
| | $n = 118$ | | X | | — | | | 2 | 0.1 | 0.16 |
| | 12 models | | X | X | — | | | 3 | 1.0 | 0.10 |
| | | | X | | — | | X | 3 | 1.1 | 0.09 |
| | | | X | X | — | | X | 4 | 1.1 | 0.09 |
| | | | | | — | | X | 2 | 1.3 | 0.09 |
| | | | X | | — | X | | 3 | 1.6 | 0.07 |
| | | | X | | — | | | 2 | 1.6 | 0.07 |
| | $\beta$ | 0.66 | −0.036 | 0.00071 | — | 0.39 | 2.14 | | | |
| | s.e. | 0.82 | 0.044 | 0.00059 | — | 0.56 | 1.99 | | | |
| | RVI | | 0.64 | 0.28 | — | 0.30 | 0.38 | | | |
| male | age (1–7) | X | | | | | | 1 | 0 | 0.22 |
| | $n = 74$ | | | | X | | | 2 | 1.6 | 0.10 |
| | | | | | | | X$^a$ | 2 | 1.8 | 0.09 |
| | 24 models | | X | | | | | 2 | 1.8 | 0.09 |
| | | | | | | X | | 2 | 1.8 | 0.09 |
| | | | X | X | | | | 3 | 3.4 | 0.04 |
| | $\beta$ | 0.85 | 0.0015 | | −0.09 | 0.07 | 0.37 | | | |
| | s.e. | 0.09 | 0.0020 | | 0.08 | 0.10 | 0.42 | | | |
| | RVI | | 0.37 | | 0.30 | 0.28 | 0.30 | | | |
| | later/earlier | | X | X | — | | | 3 | 0 | 0.29 |
| | $n = 131$ | | X | X | — | | X | 4 | 1.7 | 0.13 |
| | 12 models | X | | | — | | | 1 | 2.0 | 0.11 |
| | | | X | | — | | | 2 | 2.1 | 0.10 |
| | | | | | — | | X | 2 | 3.1 | 0.06 |
| | | | | | — | X | | 2 | 4.0 | 0.04 |
| | $\beta$ | 1.14 | −0.10 | 0.0010 | — | | −1.28 | | | |
| | s.e. | 1.07 | 0.05 | 0.00052 | — | | 1.87 | | | |
| | RVI | | 0.77 | 0.56 | — | | 0.32 | | | |

$^a$F was correlated with population size (Spearman correlation; $\rho = -0.23$, $p = 0.04$).

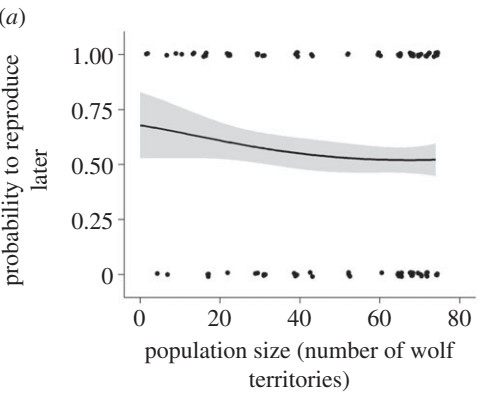 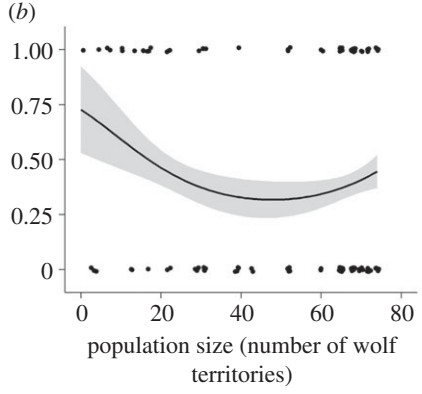

**Figure 3.** Probability to reproduce later in relation to population size (number of wolf territories) for (a) female, and (b) male wolves in Scandinavia (1988–2018; $n_{\text{females}} = 118$, $n_{\text{males}} = 131$). The lines indicate the fitted values, with associated standard errors, from the model-averaged estimates (table 1). Partner experience was held constant at 'no' and inbreeding coefficient at the mean. Dots represent the observed values and for visual purposes overlapping dots are separated.

could explain the lower probability for males to reproduce later between 2007 and 2012. Moreover, wolves less affected by inbreeding during this period were likely faster in establishing a territory and finding a partner, thus explaining why the inbreeding coefficient was included among the highest ranking models for females.

The availability of resources and inbreeding may affect the physical conditions in terms of body mass which in turn may affect age at first reproduction, with larger females reproducing earlier than smaller ones [9,17,63]. Since most Scandinavian wolves have a surplus of food [64], it is unlikely that this is a direct factor explaining body condition. An effect of inbreeding is however possible as it has been found in wolves before [49,65]. During conditions with surplus of food, the presence of helpers will be beneficial for the physical development of juveniles [22] and thereby affect age at first reproduction. In this study, we found a negative relation between primiparity and age at first reproduction for males. For individuals with multiparous origin, we lack detailed information about the presence of older siblings or other helpers/competitors in the pack during their first year of life.

The breeding range of wolves in Scandinavia has not expanded in a comparable way to the increase in population size and may have resulted in increased competition for space. Limited space for territories may explain why the probability to reproduce later increased with population density for males during the last part of the study period. That a similar positive density dependence was not observed among females may be explained by females dispersing shorter distances than males [66] and high mortality outside the wolf breeding range [67]. The Swedish and Norwegian wolf management restricts the breeding range of wolves to the southeastern part of Scandinavia [42]. Further north and west, dispersing or newly established wolves are legally culled to avoid wolf recolonization and minimize depredation of semi-domestic reindeer and free-ranging sheep by wolves. Recolonization of southern Sweden is also hampered by legal harvest and illegal hunting as well as road mortality [67]. With shorter dispersal distances females usually stay within the breeding range and, due to high turnover rates [43], find a place to settle, while males more frequently leave the breeding range. This may affect the age structure among single wolves within the breeding range with older males and younger females that stay and wait for mating opportunities.

The parent–offspring reproductions all occurred in the off-spring's natal territory during the entire study period, but nearly half of the cases occurred during the last two years of the study (2017–2018). In other populations of social species, incestuous reproductions mainly occur between individuals without early life exposure to each other [48]. Still, considering the high turnover of territorial individuals in Scandinavia [43], incestuous reproductions were rare. Age at first reproduction for the incestuous reproductions was low with six of 11 individuals reproducing at the age of one or two. Additionally, four individuals could also have reproduced earlier as the range age estimate included the age of 2 (2–3 and 2–5). The exception was a 4 year old male reproducing incestuously the first time in 1991, when the population only comprised two territories.

Life-history traits play a key role in assessing the conservation status and guiding management actions for small populations. Age at first reproduction, together with the reproductive potential (i.e. number of offspring) and body size predicts extinction risk for mammal species with high accuracy [6] and has been shown to have a strong effect on growth rate and population projections [68]. Moreover, age at first reproduction is an important component of generation time [69], to which both genetic and ecological processes best scale in their effect on the extinction risk of populations [70,71]. Understanding the effects of the spatial and temporal patterns of age at first reproduction on generation time may assist for the timing of population viability estimates [70] as well as explaining the evolutionary processes, such as the time of speciation or population divergence events [72].

False assumptions of age at first reproduction and its effect on generation time may lead to over- or underestimation of population growth rates and the rate at which the genetic variation is lost due to genetic drift and inbreeding. The use of more realistic population models, including varying fecundity rates, also avoids overharvest [5]. Licensed harvest aiming to control wolf population size in Scandinavia (independently conducted in Norway and Sweden on top of protective culling to reduce livestock losses or remove bold wolves) has, for example, been conducted almost every year since 2005, when the population was estimated to 141–160 individuals. Our study exemplifies how a high turnover of territorial individuals caused by humans affects age at first reproduction. The overall impact of humans on large carnivore population size [23,52], distribution [24,67], genetic diversity [73] and reproductive success [74] through

legal harvest and illegal hunting highlights the need of accurate estimates of life-history traits when managing small carnivore populations in human-dominated landscapes.

Ethics. All procedures including capture, handling and VHF/GPS collaring of wolves fulfilled ethical requirements and have been approved by the Swedish Animal Experiment Ethics Board (permit no. C 281/6) and the Norwegian Experimental Animal Ethics Committee (permit no. 2014/284738-1).

Data accessibility. Data available from the Dryad Digital Repository: https://doi.org/10.5061/dryad.3n5tb2rgg [75].

Authors' contributions. C.W. secured funding, conceived and designed the study, compiled data, carried out the statistical analyses and drafted the manuscript; M.G. conceived and designed the study, compiled data, carried out statistical analyses and revised the manuscript; B.Z. carried out statistical analyses and revised the manuscript; Ø.F. was involved in the laboratory work and revised the manuscript; M.A. secured funding, was involved in the laboratory work, conceived and designed the study, compiled data and drafted the manuscript. All authors gave final approval for publication and agree to be held accountable for the work performed therein.

Competing interests. We have no competing interests.

Funding. The study was funded by FORMAS (2019-01186), Swedish Environmental Protection Agency, Norwegian Environment Agency, Swedish Association for Hunting and Wildlife Management, and Marie-Claire Cronstedts Foundation.

Acknowledgements. We thank our capture crew for capturing and handling the wolves. We thank all the people who conducted laboratory- or fieldwork or administrated the monitoring of wolves in Scandinavia during the last 40 years.

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
