## [Peer Review File · Proceedings of the Royal Society B: Biological Sciences]

Review History

RSPB-2020-0986.R0 (Original submission)

Review form: Reviewer 1 (Shannon Barber-Meyer)

Recommendation

Accept with minor revision (please list in comments)

Scientific importance: Is the manuscript an original and important contribution to its field?

Excellent

General interest: Is the paper of sufficient general interest?

Good

Quality of the paper: Is the overall quality of the paper suitable?

Excellent

Is the length of the paper justified?

Yes

Should the paper be seen by a specialist statistical reviewer?

No

Do you have any concerns about statistical analyses in this paper? If so, please specify them explicitly in your report.

No

It is a condition of publication that authors make their supporting data, code and materials available - either as supplementary material or hosted in an external repository. Please rate, if applicable, the supporting data on the following criteria.

Is it accessible?

Yes

Is it clear?

Yes

Is it adequate?

Yes

Do you have any ethical concerns with this paper?

No

Comments to the Author

Thank you for the opportunity to review this manuscript. The authors report important information regarding the first age at reproduction in wild wolves of Scandinavia from the period of early wolf recolonization through increasing population levels. The rigor and resolution of the data in this manuscript are usually quite difficult to obtain but through careful, long term monitoring and the application of cutting-edge technologies and modeling, the researchers have gleaned remarkable results.

The findings have significant implications to conservation and management of wild wolves in terms of expected population trajectories (viability analyses, etc.) and also to broader issues such as estimating dog domestication (inter-generational reproduction estimates).

This research has also raised interesting questions that should be addressed in other study areas to flesh out the broader applicability of these results regarding the influence of inbreeding, territorial establishment, and mate availability on age at first reproduction relative to maturation and nutrition. In addition to containing critical information related to wolf research and conservation / management, this manuscript is also very broadly relevant to other species in other ecological situations.

The manuscript is well-written, well-reasoned and the methodological approaches are sound. The manuscript is appropriate in terms of content / scope for the journal.

I have only a few minor suggestions detailed in the attached comment summary pdf with my suggested changes in comments that are cross referenced to the relevant parts of the manuscript. (See Appendix A)

Review form: Reviewer 2

Recommendation

Reject - article is not of sufficient interest (we will consider a transfer to another journal)

Scientific importance: Is the manuscript an original and important contribution to its field?

Marginal

General interest: Is the paper of sufficient general interest?

Marginal

Quality of the paper: Is the overall quality of the paper suitable?

Marginal

Is the length of the paper justified?

Yes

Should the paper be seen by a specialist statistical reviewer?

No

Do you have any concerns about statistical analyses in this paper? If so, please specify them explicitly in your report.

No

It is a condition of publication that authors make their supporting data, code and materials available - either as supplementary material or hosted in an external repository. Please rate, if applicable, the supporting data on the following criteria.

Is it accessible?

N/A

Is it clear?

N/A

Is it adequate?

N/A

Do you have any ethical concerns with this paper?

No

Comments to the Author

Here the authors have access to a large and interesting data set and attempt to describe age at first reproduction and investigate factors that may affect it. This is an interesting question and a great system to explore these questions. Individually most sentences are well written and make sense, but they're not linked together in a way that is maximally effective. The end result is that I think that the authors have failed to tell a compelling story throughout the manuscript. The introduction needs more detail in places and broadly just needs to make it clear to the reader what the important questions that are being asked are, why these questions are important, and what the background literature has to say about them so far. Similar problems arise in the Discussion. I found that to be mostly a restating of the results without great explanations as to what it meant for the biology of the species or the larger theoretical or practical conservation body of work. I think that this work both should and could be published somewhere, but in my opinion needs a complete re-write. See line by line comments below.

Line 22. Remove e.g., for readability

Line 23. Species scientific name?

Line 27. Units should be years no? And "of SD" seems like a typo or misformatting? Are these means or medians?

Line 28. Earlier it says you estimated age at first rep. for 452 wolves, but the sample sizes here don't reflect that.

Introduction:

Line 39. I think a little more detail about what you mean by “environmental change” is needed.

Line 41. Haven’t seen e.g., used outside of parentheses like this. Suggest rewording as it’s a bit awkward.

Line 42-48. I think a little more detail on the possible benefits of delayed reproduction. What about idea of queuing for better territories/mates rather than breeding right away? How does longevity play into this – would you expect different patterns in species that live 1-2 years vs. more?

Line 49: You just sort of jump into newly re-established populations without any context. Seems like an abrupt transition.

Line 56: I would use “negative density-dependence” rather than inverse.

Line 57: Example seems kind of out of nowhere to me. Why is this example important?

Line 61-62. I think you need a little explanation as to how age at first reproduction is tied to physiological development.

Line 84:86 – True, but need to explain to the reader why this is important, and how it ties into your argument. Also, there is a much larger literature on removal experiments including several reviews that you should also consider citing here.

Line 86-89. Not following this prediction. Is it one of your predictions?

Line 120-130. Finding your first prediction very hard to follow. Re-phrase and clarify. Methods.

Line 215. What do you mean by “interval year”?

Line 216. I’m not following the first option here. 1). Re-phrase and explain.

Line 239. This paragraph only has one sentence.

Line 275-277. I would pick either mean +/- interquartile range or mean +/- SD, whichever best describes your data with regards to its distribution.

Line 278. This paragraph only has one sentence. Is there a difference in age at first reproduction between these individuals and the rest of the population?

Discussion:

Line 308: Low compared to what?

First paragraph: This paragraph in my mind should really quickly summarize your findings and explain why they’re important. Were your hypotheses/predictions met or not?

Decision letter (RSPB-2020-0986.R0)

23-Jun-2020

Dear Dr Wikenros,

I am writing to inform you that your manuscript RSPB-2020-0986 entitled "Age at first reproduction is density-dependent in a recolonizing carnivore population" has, in its current form, been rejected for publication in Proceedings B.

We are all agreed that this is a really impressive data-set and an important question. However the reviewers and Associate Editor have recommended that substantial revisions are necessary for the manuscript to be suitable for publication. With this in mind we would be happy to consider a resubmission, provided the comments of the referees are fully addressed. However please note that this is not a provisional acceptance.

Yours sincerely,
Professor Loeske Kruuk
<mailto:proceedingsb@royalsociety.org>

Editor comment: Better title? My first reaction on reading the title was 'we know that age at first reproduction is density dependent in many systems'. It's only on reading the abstract that the interesting nature of the relationship becomes apparent. Is it possible to capture that in the title? I would also state that it's about wolves in the title.

Associate Editor
Board Member: 1
Comments to Author:

I agree with both reviewers that the data, analysis, and main conclusions are well-supported and important. I also agree with reviewer two that before this manuscript can be published, it will be crucial to explain in the introduction and discussion why this work is important for this area of ecological theory (life history evolution and density dependence etc), and also state more clearly how the findings relate the biology of wolves and their management broadly. The text needs some re-writing to make this story and it's wider implications clear.

Reviewer(s)' Comments to Author:

Referee: 1

Comments to the Author(s)

Thank you for the opportunity to review this manuscript. The authors report important information regarding the first age at reproduction in wild wolves of Scandinavia from the period of early wolf recolonization through increasing population levels. The rigor and resolution of the data in this manuscript are usually quite difficult to obtain but through careful, long term monitoring and the application of cutting-edge technologies and modeling, the researchers have gleaned remarkable results.

The findings have significant implications to conservation and management of wild wolves in terms of expected population trajectories (viability analyses, etc.) and also to broader issues such as estimating dog domestication (inter-generational reproduction estimates).

This research has also raised interesting questions that should be addressed in other study areas to flesh out the broader applicability of these results regarding the influence of inbreeding, territorial establishment, and mate availability on age at first reproduction relative to maturation and nutrition. In addition to containing critical information related to wolf research and conservation / management, this manuscript is also very broadly relevant to other species in other ecological situations.

The manuscript is well-written, well-reasoned and the methodological approaches are sound. The manuscript is appropriate in terms of content / scope for the journal.

I have only a few minor suggestions detailed in the attached comment summary pdf with my suggested changes in comments that are cross referenced to the relevant parts of the manuscript.

Referee: 2

Comments to the Author(s)

Here the authors have access to a large and interesting data set and attempt to describe age at first reproduction and investigate factors that may affect it. This is an interesting question and a great system to explore these questions. Individually most sentences are well written and make sense, but they're not linked together in a way that is maximally effective. The end result is that I think that the authors have failed to tell a compelling story throughout the manuscript. The introduction needs more detail in places and broadly just needs to make it clear to the reader what the important questions that are being asked are, why these questions are important, and what the background literature has to say about them so far. Similar problems arise in the Discussion. I found that to be mostly a restating of the results without great explanations as to what it meant for the biology of the species or the larger theoretical or practical conservation body of work. I think that this work both should and could be published somewhere, but in my opinion needs a complete re-write. See line by line comments below.

Line 22. Remove e.g., for readability

Line 23. Species scientific name?

Line 27. Units should be years no? And "of SD" seems like a typo or misformatting? Are these means or medians?

Line 28. Earlier it says you estimated age at first rep. for 452 wolves, but the sample sizes here don't reflect that.

Introduction:

Line 39. I think a little more detail about what you mean by “environmental change” is needed.

Line 41. Haven’t seen e.g., used outside of parentheses like this. Suggest rewording as it’s a bit awkward.

Line 42-48. I think a little more detail on the possible benefits of delayed reproduction. What about idea of queuing for better territories/mates rather than breeding right away? How does longevity play into this – would you expect different patterns in species that live 1-2 years vs. more?

Line 49: You just sort of jump into newly re-established populations without any context. Seems like an abrupt transition.

Line 56: I would use “negative density-dependence” rather than inverse.

Line 57: Example seems kind of out of nowhere to me. Why is this example important?

Line 61-62. I think you need a little explanation as to how age at first reproduction is tied to physiological development.

Line 84:86 – True, but need to explain to the reader why this is important, and how it ties into your argument. Also, there is a much larger literature on removal experiments including several reviews that you should also consider citing here.

Line 86-89. Not following this prediction. Is it one of your predictions?

Line 120-130. Finding your first prediction very hard to follow. Re-phrase and clarify.

Methods.

Line 215. What do you mean by “interval year”?

Line 216. I’m not following the first option here. 1). Re-phrase and explain.

Line 239. This paragraph only has one sentence.

Line 275-277. I would pick either mean +/- interquartile range or mean +/- SD, whichever best describes your data with regards to its distribution.

Line 278. This paragraph only has one sentence. Is there a difference in age at first reproduction between these individuals and the rest of the population?

Discussion:

Line 308: Low compared to what?

First paragraph: This paragraph in my mind should really quickly summarize your findings and explain why they’re important. Were your hypotheses/predictions met or not?

Author's Response to Decision Letter for (RSPB-2020-0986.R0)

See Appendix B.

RSPB-2021-0207.R0

Review form: Reviewer 1

Recommendation

Accept as is

Scientific importance: Is the manuscript an original and important contribution to its field?

Good

General interest: Is the paper of sufficient general interest?

Good

Quality of the paper: Is the overall quality of the paper suitable?

Excellent

Is the length of the paper justified?

Yes

Should the paper be seen by a specialist statistical reviewer?

No

Do you have any concerns about statistical analyses in this paper? If so, please specify them explicitly in your report.

No

It is a condition of publication that authors make their supporting data, code and materials available - either as supplementary material or hosted in an external repository. Please rate, if applicable, the supporting data on the following criteria.

Is it accessible?

No

Is it clear?

N/A

Is it adequate?

N/A

Do you have any ethical concerns with this paper?

No

Comments to the Author

Please see the attached file. (See Appendix C)

Decision letter (RSPB-2021-0207.R0)

22-Feb-2021

Dear Dr Wikenros and co-authors,

I am pleased to inform you that your manuscript RSPB-2021-0207 entitled "Age at first reproduction in wolves: different patterns of density dependence for females and males" has been accepted for publication in Proceedings B.

The referee and Associate Editor have recommended publication of the manuscript as it is, but have noted a problem with the Dryad reference to archived data. Therefore, I invite you to check over the manuscript and also address this issue. Because the schedule for publication is very tight, it is a condition of publication that you submit the revised version of your manuscript within 7 days. If you do not think you will be able to meet this date please let us know.

In order to ensure effective and robust dissemination and appropriate credit to authors the dataset(s) used should be fully cited. To ensure archived data are available to readers, authors

should include a 'data accessibility' section immediately after the acknowledgements section. This should list the database and accession number for all data from the article that has been made publicly available, for instance:

[http://datadryad.org/submit?journalID=RSPB&manu=\(Document not available\)](http://datadryad.org/submit?journalID=RSPB&manu=(Document+not+available)) which will take you to your unique entry in the Dryad repository. If you have already submitted your data to dryad you can make any necessary revisions to your dataset by following the above link. Please see <https://royalsociety.org/journals/ethics-policies/data-sharing-mining/> for more details.

Sincerely,
Professor Loeske Kruuk
<mailto:proceedingsb@royalsociety.org>

Associate Editor
Comments to Author:

I agree with the reviewer that this will be a nice contribution, the authors have now addressed all of the comments thoroughly, and have included helpful context of life history theory and wolf biology for management.

Reviewer(s)' Comments to Author:

Referee: 1

Comments to the Author(s).
Please see the attached file.

Author's Response to Decision Letter for (RSPB-2021-0207.R0)

See Appendix D.

Decision letter (RSPB-2021-0207.R1)

09-Mar-2021

Dear Dr Wikenros

I am pleased to inform you that your manuscript entitled "Age at first reproduction in wolves: different patterns of density dependence for females and males" has been accepted for publication in Proceedings B.

Data Accessibility section

Open Access

Paper charges

Sincerely,

Proceedings B

Appendix A

17 **Abstract**

Age at first reproduction constitutes a key demographic trait in animals and is
evolutionary shaped by fitness benefits and costs of delayed versus early
reproduction. The understanding of how environmental change affects age at first
reproduction is crucial for conservation and management of threatened species
because of its demographic effects on e.g., population growth and generation time.
We estimated age at first reproduction for 452 reproducing wolves during 40 years
including the recolonization phase of the Scandinavian wolf population and examined
how the variation among individuals was affected by sex, population size (from 1 to
74 packs), primiparous or multiparous origin, reproductive experience of the partner,
and inbreeding. Average age at first reproduction was 2.82 ± 1.05 of SD for females
($n = 60$) and 2.39 ± 0.84 for males ($n = 74$), and ranged between 1 and 8-10 years of
age. Female age at first reproduction decreased with increasing population size. The
probability for males to reproduce late first decreased, reaching its minimum when the
number of territories approached 40 to 60, and then increased with increasing
population size. Other factors had weak correlation with age at first reproduction. The
results support an Allee effect during wolf establishment and that Scandinavian
wolves have been more limited by space and/or mates than conditions affecting
maturation.

Summary of Comments on

Page: 4

-
- Author: sbarber-meyer Subject: Sticky Note Date: 5/15/2020 3:49:31 PM
If possible, can you include a brief mention in the abstract of how you are defining "first reproduction" - evidence of pregnancy (placental scars), evidence of pups surviving to be nursed, or successfully recruiting pups into popn, or? It seems that based on the genetics and capture work described below that this is not age of first pregnancy but age of first successful whelping and pup survival to at least age of capture/collaring?
-
- Author: sbarber-meyer Subject: Highlight Date: 5/15/2020 9:26:07 AM
Late relative to?

[41,42] and body condition [43], where less developed individuals in poor condition
may be expected to stay longer with their parents and thus delay their reproduction.

With data from the recently recolonized and fairly isolated population of
wolves in Scandinavia we gain better understanding of the relative importance of both
environmental and genetic factors influencing age at first reproduction and how this
change during its re-establishment. We estimated the age at first reproduction for 452
reproducing wolves and examined how the variation among individuals was affected
by population size, primiparous or multiparous origin, reproductive experience of the
partner, and inbreeding coefficient (F) as well as classification of individuals as first-,
second-, or third-generation immigrant descendants or resident origin. We predicted
age at first reproduction to; 1) be quadratically related to population size with older
age at small sizes due to an Allee effect and at large sizes due to competition for food
and space, but younger age at intermediate population sizes due to increased
availability of mates; 2) be lower for wolves with multiparous than primiparous origin
due to the benefits of parental learning and the potential access to more food from
helpers and/or more experienced parents; and 3) be lower for those having a partner
with previous breeding experience given that individuals prefer to pair up with an
established mate rather than settling in a new unoccupied territory with an
unexperienced mate. While testing these predictions for females and males separately
we 4) also included the effect of inbreeding with the prediction that it delays the age
at first reproduction.

**Methods**

*Study area*

The study was conducted on the Scandinavian Peninsula (south-eastern Norway and
south-central Sweden, hereafter referred to as Scandinavia) in the boreal forest zone.
The climate is continental and snow covers the ground mainly during December to
March. The forests are mostly composed of Norway spruce (*Picea abies*), Scots pine
(*Pinus sylvestris*) and some deciduous species, mostly aspen (*Populus tremula*) and
birch (*Betula* spp.). Extensive commercial logging and forest management practices
have resulted in a network of gravel roads with an average density of 0.9 km/km²
within wolf territories [44]. Human density was 24 humans per km² in 2016 in
Sweden [45] and 17 humans per km² in 2017 in Norway [46], but large areas within
the wolf population range have less than 1 human per km².

The wolf was declared functionally extinct in Scandinavia in 1966. In 1983,
two wolves from the Finnish-Russian population reproduced in a cross-border
territory of Sweden and Norway, and thereby founded the current Scandinavian
population [34,36]. Since then, pups were born every year, except for 1986, and
during 1987-1990 the population was maintained by incestuous reproductions
resulting in inbreeding [36]. In 1991, wolves reproduced for the first time in two
territories and the male in the second territory was a third Finnish-Russian immigrant
[35]. By then, the wolf population started to increase in numbers and expand their
breeding range in Scandinavia (Figure 1). Later, five more immigrants have
successfully reproduced, two males in 2008, one translocated pair in 2013, one male
in 2016, and one female in 2017 [35,36,39,47]. License harvest of wolves occurred
during January-February in 2010, 2011, 2015-2018 in Sweden, during 2005, 2007-
2009, 2011-2018 in Norway, and protective culling occurred annually in both
countries. Poaching was the single most common mortality cause of Scandinavian
wolves during 2000-2017 [48]. Pair dissolution rate among Scandinavian wolves have

Page: 9

Author: sbarber-meyer Subject: Highlight Date: 5/15/2020 3:39:18 PM
I think this should be "rates".

All procedures including capture, handling and VHF/GPS collaring of wolves [56,57],
performed by the Scandinavian wolf research project SKANDULV and occasionally
by management authorities in Norway and Sweden, fulfilled ethical requirements and
have been approved by the Swedish Animal Experiment Ethics Board (Permit
Number: C 281/6) and the Norwegian Experimental Animal Ethics Committee
(Permit Number 2014/284738-1). Samples were derived both from dead and live-
captured wolves, and from non-invasive samples collected during the annual
Norwegian-Swedish wolf monitoring while snow-tracking [39]. Sex was determined
either from morphological sexing of dead or captured individuals, or from DNA-
analysis (see Supplement for further details).

To determine parental identities and to reconstruct the pedigree, we used a
two-step process based on microsatellite genotypes [39] and field observations
[36,39]. ~~At first hand,~~ individual parents were determined by genetic exclusion of
putative parents, i.e., pairs known to have scent-marked in the same territory. If all
putative parents could be excluded assuming no more than two mismatches, we used
parental assignment in CERVUS v3.0 from the entire database of individuals
identified between 1983 and 2017 [39]. Of 452 breeding individuals we were able to
determine the population of origin and the parental identities of Scandinavian born
individuals to previous genotyped individuals in 408 cases (90%). In 43 cases the
parental genotypes could be reconstructed to such a degree that the grand-parents
could be identified. That leaves one individual (< 1%) where the genealogy could not
be reconstructed. Based on the reconstructed pedigree, we calculated the inbreeding
coefficient (F) of the breeding individual using CFC v1.0 [58] and classified their
relationship to immigrants as F1, F2, or F3 (first-, second-, or third-generation
immigrant descendants, respectively) or native inbred, i.e., no close relationship with

Page: 11

Author: sbarber-meyer	Subject: Inserted Text	Date: 5/15/2020 3:43:46 PM
------------------------	----------------------------

Firstly,

immigrants [39]. Some individuals were excluded from the analyses, including one
individual with unknown F and 11 individuals that reproduced the first time with a
parent and thus may be influenced by other factors than the ones included in the
study.

*Age determination*

We used three sources of information to determine the year, or interval year, of first
reproduction of individuals including: 1) the first year of parental reproduction, that
gives the year of birth when the offspring was identified less than 12 months after the
reproductive event or when the parental pair only reproduced one year, 2) the
individual was identified as a pup at the den, or, 3) the individual was identified and
aged as a juvenile when captured during its first winter. A handled wolf was aged to <
1 year using several combined methods, including no or little visible tooth wear,
puppy fur, presence of a juvenile-specific growth zone on the front leg (tibia) which
disappears before 1 ½ year old [28] or if in doubt by post-collaring typical
Scandinavian wolf pup behavior, e.g., no killing of ungulates, but dependent on
feeding at prey killed by their parents or kinship yearlings [59]. Age at first
reproduction was unknown for 9 individuals consisting of immigrants or individuals
with unknown parents. Since the vast majority of individuals that bred for their first
time were either 2 years or 3 years of age, we also investigated the determinants of
first-time breeding using two discrete age classes, including 1-2 years of age (defined
as early first reproduction) versus ≥ 3 years of age (defined as late first reproduction).

*Primiparous or multiparous origin and partner experience*

Page: 12

Author: sbarber-meyer Subject: Highlight Date: 5/18/2020 10:37:17 AM

This is semantics - but "late" seems to reference an absolute (but 3 yrs in other studies would not be considered late) - suggestion to refer to these categories throughout the manuscript as "earlier" and "later" rather than "early" and "late" (or by just directly age classes such as ≤ 2 and > 2) because earlier and later imply a reference to these data rather than perhaps an external absolute. I hope that isn't too confusing - what I'm trying to get at is I would not consider 3 yr particularly "late" - but it is "later" than the other category of 1-2 yr you are considering.

with pair identity as a random factor. Year of first time reproduction was used as
 random factor, to account for autocorrelation between years. For both analyses, we
 compared candidate models using the sample-size corrected Akaike Information
 Criterion (AIC_c) and AIC weights (w_i) from the 'MuMIn' package [62] in R. Models
 with $\Delta AIC \leq 2$ were used to generate model-averaged parameter estimates [63]. We
 used AIC weights on model set with $\Delta AIC \leq 2$ to generate Relative Variable
 Importance weights (RVI) for each explanatory variable.

Results

Age at first reproduction was given for 134 individuals, whereas for 297 individuals
 age at first reproduction was estimated with an uncertainty of two ($n = 120$), three (n
 $= 99$), or 4-9 years ($n = 78$) (Supplement, Figure S1) and 115 classified as early or late
 reproducers. Of the 134 wolves with exact age 1% reproduced at one year of age (two
 males), 59% at two years, 28% at three years, and 12% at four years or older. Of the
 249 individuals (including the ones with exact years) which could be classified into
 early and late reproducers, 52% reproduced at the age of one or two years, and 48%
 reproduced at three years or later. The oldest age at first reproduction was 8-10 years
 for two females during 2001. Median age at first reproduction was 3 (range 2-7) and 2
 (range 1-7), with average of 2.82 ± 1.05 SD and 2.39 ± 0.84 for females ($n = 60$) and
 males ($n = 74$), respectively.

Six females and five males reproduced the first time with their own parent.

Age at first reproduction for those individuals (sex and years of first reproduction in
 subscript) was 1 ($n_{\text{♀}2018} = 1$), 1-2 ($n_{\text{♀}2002} = 1$, $n_{\text{♀}2006} = 1$), 2 ($n_{\text{♀}2004,2015} = 2$, $n_{\text{♀}2017} = 1$),
 2-3 ($n_{\text{♀}2018} = 1$, $n_{\text{♀}2013} = 1$), 4 ($n_{\text{♂}1991} = 1$), and 2-5 ($n_{\text{♀}2017} = 2$) years.

Page: 14

Author: sbarber-meyer Subject: Sticky Note Date: 5/18/2020 9:36:16 AM

Suggestion to analyze results with respect to potential overdispersion and include this in results (e.g., in GLM the residual deviance divided by the residual degrees of freedom demonstrates whether overdispersion was potentially an issue compared to the dispersion parameter of 1 for the Poisson family in GLM, then sometimes one can inflate c-hat to look at possible changes in model selection results / robustness of selection to overdispersion, if overdispersion was an issue). I don't know how the dispersion equation changes for GLMM but it is my understanding that under/overdispersion can still be assessed (and families changed if a big issue and / or dispersion parameter can be modeled). I suspect this was not an issue - but best, if possible, to demonstrate that it was not an issue using data (results).

Author: sbarber-meyer Subject: Sticky Note Date: 5/18/2020 9:43:09 AM

Wow these are some really interesting findings regarding wolves that don't breed until 7 yrs plus. Just living to 7 years in the wild is an accomplishment - then to think they then start breeding at 7 yrs+ - wow - wonder if this could be an ecologically stable strategy (ESS) in game theory that might be an alternative to breeding at 2-4 years of age (if they have different cost and benefits as evidenced by average survival and reproduction).

*Age at first reproduction as continuous response*

In females, age at first reproduction decreased with increasing population size (Figure  2a). The inbreeding coefficient F was included in the top models, but the standard
error around the estimate of the effect included zero indicating only a weak positive
correlation between inbreeding coefficient and age at first reproduction (Table 1). The
immigrant relationship variable was not retained in the top models (Supplement,
Table S2).

For males, the top models of continuous age at first reproduction included
weak relationships with population size, primiparity origin, partner experience, and F
(all confidence intervals of the estimates included zero), and the highest ranking
model was the intercept model (Figure 2b, Table 1, Supplement, Table S2).

*Late or early first reproduction*

In females, the top models for the probability to reproduce late included population
size as linear term and partner experience, but those relationships were weak (the
confidence interval of the coefficients included zero), and the intercept model was the
highest ranking model (Table 1, Figure 3a, Supplement, Table S2).

In males, the best models contained population size as a polynomial term, with
a U-shaped relationship between the probability to reproduce late and population size
(Table 1, Figure 3b). The inbreeding coefficient was also contained in the top models,
but the confidence interval around the estimate of the effect included zero indicating a
weak relationship (Table 1). The immigrant relationship variable was not retained in
the top models (Supplement, Table S2).

**Discussion**

Page: 15

Author: sbarber-meyer Subject: Sticky Note Date: 5/18/2020 10:38:47 AM

Do you think that your observed population levels reflect the upper level of saturation? If not, I wonder if the population became even more saturated if female age at first repro might again increase as you hypothesized. Could it be that the population levels you have observed so far are restricted to low and intermediate ?

In the Scandinavian wolf population, age at first reproduction was low with an
average of 2.82 and 2.39 years of age for females and males, respectively, and 52-
60% (depending on age estimate accuracy) of the individuals reproduced for the first
time at the age of 1-2 years. Wolves may breed as early as at one year of age [25,26],
although ~~rare~~ in Scandinavia, three males reproduced at one year of age, two non-
incestuously and one incestuously, but all in the vicinity to or within their natal
territory. Breeding at one year of age is thus a rare event in Scandinavia and so far
only confirmed for males. In North American populations, wolves mostly start
breeding at two years of age [27–30], but in some areas, female wolves do not
normally breed until four years of age [31,32]. In our study, age at first reproduction
ranged from one to 8-10 years, and in the following, we discuss factors that explain
some of the observed individual variation, but also factors that we initially considered
important for age at first reproduction, but turned out to have weak effects in the
analyses.

Population size, in terms of number of territories with ≥ 2 resident wolves
during winter, was the most prevalent and strongest factor affecting age at first
reproduction in Scandinavian wolves, for both sexes. In the early phase of the
population's history, we found indications that the population was subject to an Allee
effect in both males and females. The treatment of age at first reproduction as either a
continuous variable (years of age) or categorical variable (late versus early first
reproduction) differed in its association with population size in females and males.
Female age decreased with increasing population size, but the probability of late
reproduction was only weakly related to population size. For males, the probability to
reproduce late had a U-shaped relationship with population size. Generally, males
showed a lower variation in age at first reproduction than females, with proportionally

more males being either two years or three years, indicating that a binomial model
better explains the reproductive age of males than a Poisson model. In contrast, a
Poisson model is likely better in explaining reproductive age for females than males
since there was higher variation in age of late reproducers among females, variation
that is lost when treating the data binomially.

Both females and males reproduced late during the early phase of the
population development. The breeding range of wolves in Scandinavia has not
expanded in a comparable way to the increase in population size and likely resulted in
increased competition for space. This may explain the quadratic effect of age at first
reproduction for males with higher probability to reproduce late during the last part of
the study period. The mechanism behind why this pattern was not observed for
females remains unknown but may be a result of different pattern of territory
establishment where female establish territories early and males establish when
pairing with a female that already hold a territory.

In Wisconsin and Michigan, USA, Stenglein and Van Deelen [64] estimated
that a population crossed the Allee threshold at roughly 20 wolves in four to five
packs. The decreasing age of reproduction and probability of late reproduction found
in this study could partly be due to an Allee effect during recolonization of the
Scandinavian Peninsula. Still, for males the turning point between positive and
negative density dependence of probability for late reproduction was as much as 40-
60 territories and it is likely that there are other factors apart from an Allee effect
explaining the yearly variation in age at first reproduction.

The period of decreasing age at first reproduction coincides with an increasing
turn-over of territorial individuals [48,49]. Partner experience was included in the best
models for both females and males even though the effect was weak, indicating that

Page: 17

Author: sbarber-meyer Subject: Sticky Note Date: 5/18/2020 10:08:17 AM

If this sex-age difference in territory establishment was documented in your study, I suggest to note that again here to make it clear that this is more than hypothetical.

the loss of partners gives young adults the opportunity to find a partner to a higher
degree than older individuals. Populations that experience high turnover of territorial
individuals due to anthropogenic hunting are likely to experience a higher variation in
age at first reproduction due to the varying access to resources for surviving non-
territorial individuals [65]. Hunting may also have a negative impact on longevity of
pair relations and increased access on unpaired territorial partners [49]. The
disappearance rate of territorial pairs in the Scandinavian wolf population increased
from 0.09 during the period 2000-2009 to 0.21 during the period 2010-2016, where
the increased disappearance rate during the latter time period was mainly explained by
poaching [48]. Despite this high disappearance rate of territorial pairs, the number of
territories increased from 65 to 74 during the last years of the study period (2012-
2017). The high turn-over of territorial pairs may be one of the reasons for the low
age at first reproduction when territorial individuals are quickly replaced.

During the same period as the probability for males to reproduce late was at its
lowest (2007-2010 where the number of territories was 39-60), two immigrant wolves
(males) reproduced three years in a row, after a 17-year period without effective
immigration. Average annual population growth before this event (2002-2007) was
13% and the corresponding number for the 2007-2012 period was 27% [39]. It is
possible that this genetic rescue event increased the availability of mates, which partly
could explain the lower probability for males to reproduce late between 2007 and
2012. With increasing variation in inbreeding during this period, it may well be that
wolves less affected by inbreeding ~~was~~ faster in establishing a territory and finding a
partner. This may explain why the inbreeding coefficient was included among the
highest ranking models for females and males, even though the effect appeared
relatively weak.

Appendix B

Dear Prof. Kruuk,

Please see below each comment and our responses (in bold). Line numbers refer to those in the newly revised manuscript with marked changes.

With best regards,

Camilla Wikenros

Editor comment: Better title? My first reaction on reading the title was 'we know that age at first reproduction is density dependent in many systems'. It's only on reading the abstract that the interesting nature of the relationship becomes apparent. Is it possible to capture that in the title? I would also state that it's about wolves in the title.

Our response: We have changed the title to “Age at first reproduction in wolves: different patterns of density dependence for females and males”.

Associate Editor

Board Member: 1

Comments to Author:

I agree with both reviewers that the data, analysis, and main conclusions are well-supported and important. I also agree with reviewer two that before this manuscript can be published, it will be crucial to explain in the introduction and discussion why this work is important for this area of ecological theory (life history evolution and density dependence etc), and also state more clearly how the findings relate the biology of wolves and their management broadly. The text needs some re-writing to make this story and it's wider implications clear.

Our response: Thank you very much. We have added the importance of increased knowledge regarding age at first reproduction in conservation (population viability analyses) and management (harvest models) of wolves and particularly for small, reintroduced or recolonizing, populations. This is done both in the Introduction (lines 169-171) and in the Discussion (lines 569-597). We have extended the part regarding density-dependent age at first reproduction with focus on small populations in the Introduction (lines 59-76).

Reviewer(s)' Comments to Author:

Referee: 1

Comments to the Author(s)

Thank you for the opportunity to review this manuscript. The authors report important information regarding the first age at reproduction in wild wolves of Scandinavia from the period of early wolf recolonization through increasing population levels. The rigor and resolution of the data in this manuscript are usually quite difficult to obtain but through careful, long term monitoring and the application of cutting-edge technologies and modeling, the researchers have gleaned remarkable results.

The findings have significant implications to conservation and management of wild wolves in terms of expected population trajectories (viability analyses, etc.) and also to broader issues such as estimating dog domestication (inter-generational reproduction estimates).

This research has also raised interesting questions that should be addressed in other study areas to flesh out the broader applicability of these results regarding the influence of inbreeding, territorial establishment, and mate availability on age at first reproduction relative to maturation and nutrition. In addition to containing critical information related to wolf research and conservation / management, this manuscript is also very broadly relevant to other species in other ecological situations.

The manuscript is well-written, well-reasoned and the methodological approaches are sound. The manuscript is appropriate in terms of content / scope for the journal.

I have only a few minor suggestions detailed in the attached comment summary pdf with my suggested changes in comments that are cross referenced to the relevant parts of the manuscript.

Our response: Thank you very much for your positive comments. We have added the implications to conservation and management in the introduction (lines 169-171) and discussion (lines 569-597).

Comment summary pdf:

Line 18 and 30. If possible, can you include a brief mention in the abstract of how you are defining "first reproduction" - evidence of pregnancy (placental scars), evidence of pups surviving to be nursed, or successfully recruiting pups into popn, or? It seems that based on the genetics and capture work described below that this is not age of first pregnancy but age of first successful whelping and pup survival to at least age of capture/collaring?

Our response: We have added that it is age at first successful reproduction with pups surviving at least three weeks of age (lines 26, 147, 215-218).

Line 115: Is this supposed to be "changed"?

Our response: We have revised the text to "changed" (line 146).

Line 158. I think this should be "rates".

Our response: This text is moved to the introduction and we have revised the text to "rates" (line 125).

Line 196. Replace At first hand with Firstly.

Our response: We have revised the text to "Firstly" (line 237).

Line 230. This is semantics - but "late" seems to reference an absolute (but 3 yrs in other studies would not be considered late) - suggestion to refer to these categories throughout the manuscript as "earlier" and "later" rather than "early" and "late" (or by just directly age classes such as <2 and >2) because earlier and later imply a reference to these data rather than perhaps an external absolute. I hope that isn't too confusing - what I'm trying to get at is I would not consider 3 yr particularly "late" - but it is "later" than the other category of 1-2 yr you are considering.

Our response: We have changed to earlier and later in the manuscript.

Line 260. Suggestion to analyze results with respect to potential overdispersion and include this in results (e.g., in GLM the residual deviance divided by the residual degrees of freedom demonstrates whether overdispersion was potentially an issue compared to the dispersion parameter of 1 for the Poisson family in GLM, then sometimes one can inflate $c\text{-hat}$ to look at possible changes in model selection results / robustness of selection to overdispersion, if overdispersion was an issue). I don't know how the dispersion equation changes for GLMM but it is my understanding that under/overdispersion can still be assessed (and families changed if a big issue and / or dispersion parameter can be modeled). I suspect this was not an issue - but best, if possible, to demonstrate that it was not an issue using data (results).

Our response: Thank you for the reminder. We have checked for overdispersion and there was a need to adjust for underdispersion. We have redone the analyses using a quasipoisson distribution and quasi-AIC-selection when testing age at first reproduction as a continuous variable (lines 290-312).

Line 274. Wow these are some really interesting findings regarding wolves that don't breed until 7 yrs plus. Just living to 7 years in the wild is an accomplishment - then to think they then start breeding at 7 yrs+ - wow - wonder if this could be an ecologically stable strategy (ESS) in game theory that might be an alternative to breeding at 2-4 years of age (if they have different cost and benefits as evidenced by average survival and reproduction).

Our response: Whether this could be considered an ESS or not is unfortunately beyond the scope and the data of this study. Five wolves were 5 years or older when they reproduced the first time and three of the cases consisted of females breeding in 1992, 1998, and 1999 at the age of 5, 6 and 7. Population size at this time was small and these late-breeding females contribute to our conclusion about density-dependent effects on age of first reproduction and possible Allee effects in the early history of the population. One male reproduced at the age of 7 in 2017, he was in a pair with a female between 2011 and 2015 and she was likely sterile as the male later paired with another female and had pups in 2017. Lastly, one female reproduced at the age of 5 in 2009 and we do not know what caused her to reproduce late.

Line 284. Do you think that your observed population levels reflect the upper level of saturation? If not, I wonder if the population became even more saturated if female age at first repro might again increase as you hypothesized. Could it be that the population levels you have observed so far are restricted to low and intermediate ?

Our response: There is no documentation that the Scandinavian wolf population is saturated with respect to availability of food and/or space for new territories. Still, the population is not distributed evenly throughout Sweden and Norway but rather clumped in a core area in south-central Scandinavia. We have added text about this in the Discussion (lines 458-477).

Line 312. I think this should be either "rarely" or "although this is rare".

Our response: We have changed to "although this is rare" (line 375).

Line 345. If this sex-age difference in territory establishment was documented in your study, I suggest to note that again here to make it clear that this is more than hypothetical.

Our response: We have extended this line of reasoning in line with the previous comment and refer to differences in dispersal distances between females and males (lines 458-477).

Line 379. I think this should be "were"

Our response: We have changed to “were” (line 439).

Referee: 2

Comments to the Author(s)

Here the authors have access to a large and interesting data set and attempt to describe age at first reproduction and investigate factors that may affect it. This is an interesting question and a great system to explore these questions. Individually most sentences are well written and make sense, but they're not linked together in a way that is maximally effective. The end result is that I think that the authors have failed to tell a compelling story throughout the manuscript. The introduction needs more detail in places and broadly just needs to make it clear to the reader what the important questions that are being asked are, why these questions are important, and what the background literature has to say about them so far. Similar problems arise in the Discussion. I found that to be mostly a restating of the results without great explanations as to what it meant for the biology of the species or the larger theoretical or practical conservation body of work. I think that this work both should and could be published somewhere, but in my opinion needs a complete re-write. See line by line comments below.

Our response: Thank you, we agree and have clarified the introduction in terms of adding relevant literature, rewriting our research questions and point out the importance of this research. We have restructured, rewritten, and put our study in a broader context both in the Introduction and in the Discussion.

Line 22. Remove e.g., for readability

Our response: We have removed e.g. (line 23).

Line 23. Species scientific name?

Our response: We have added this (line 24).

Line 27. Units should be years no? And “of SD” seems like a typo or misformatting? Are these means or medians?

Our response: We have added years, removed “of” and changed to medians (lines 30-31).

Line 28. Earlier it says you estimated age at first rep. for 452 wolves, but the sample sizes here don't reflect that.

Our response: We removed 452 as this reflects the total sample size and analyses are conducted on different sample sizes (lines 26 and 147). We have added sample size for the analyses of later or earlier first reproduction (line 32).

Introduction:

Line 39. I think a little more detail about what you mean by “environmental change” is needed.

Our response: We have removed environmental change and modified this sentence (lines 47-50). We have also changed this in the abstract (line 21).

Line 41. Haven't seen e.g., used outside of parentheses like this. Suggest rewording as it's a bit awkward.

Our response: We have removed e.g. (line 49).

Line 42-48. I think a little more detail on the possible benefits of delayed reproduction. What about idea of queuing for better territories/mates rather than breeding right away? How does longevity play into this – would you expect different patterns in species that live 1-2 years vs. more?

Our response: We have added details regarding delayed reproduction (lines 52-56) but we have not added longevity. We have considered to add that population viability is dependent on population density and growth rate especially for large, long-lived species (Soulé ME. 1987 Viable populations for conservation. Cambridge: Cambridge University Press.) and can add this and more details if requested.

Line 49: You just sort of jump into newly re-established populations without any context. Seems like an abrupt transition.

Our response: We agree and have added a few sentences to put the context for age at first reproduction in relation to newly established populations (lines 59-64).

Line 56: I would use “negative density-dependence” rather than inverse.

Our response: We have rewritten this sentence and deleted inverse (line 70-74).

Line 57: Example seems kind of out of nowhere to me. Why is this example important?

Our response: We have removed this example (lines 76-79)

Line 61-62. I think you need a little explanation as to how age at first reproduction is tied to physiological development.

Our response: We have replaced physiological development with body weight (lines 80-81).

Line 84:86 – True, but need to explain to the reader why this is important, and how it ties into your argument. Also, there is a much larger literature on removal experiments including several reviews that you should also consider citing here.

Our response: We have added that genetic differences remain to be tested (lines 109-110). We have moved the sentence regarding that it is mainly first time breeders that replace territorial individuals (line 128-130) to the section dealing with the high pair dissolution rates among Scandinavian wolves (the latter is moved from the description of the Scandinavian wolf population in the study area). We have modified the sentence about removal experiments in birds and added one reference (line 127-129).

Line 86-89. Not following this prediction. Is it one of your predictions?

Our response: We have removed this sentence.

Line 120-130. Finding your first prediction very hard to follow. Re-phrase and clarify.

Our response: We have rephrased the first and the last of the predictions and modified the rest of the predictions (lines 152-166).

Methods.

Line 215. What do you mean by “interval year”?

Our response: We have replaced this with “a range of years” (lines 258-259).

Line 216. I'm not following the first option here. 1). Re-phrase and explain.

Our response: We have re-phrased this to "...1) year of birth was given to offspring observed within the first year that the parents reproduced and to offspring with parents that only reproduced a single year..." (lines 259-261).

Line 239. This paragraph only has one sentence.

Our response: We have added this sentence to the previous paragraph (line 284).

Line 275-277. I would pick either mean +- interquartile range or mean +- SD, whichever best describes your data with regards to its distribution.

Our response: We have kept median and range due to the quasipoisson distribution (lines 30-31, 329-330).

Line 278. This paragraph only has one sentence. Is there a difference in age at first reproduction between these individuals and the rest of the population?

Our response: The sample size do not allow for a comparison with the incestuous pairs and the rest of the population. We moved this information to the methods (lines 253-255).

Discussion:

Line 308: Low compared to what?

Our response: We have removed low in this sentence (line 371) and in the following text we compare with North American wolf populations.

First paragraph: This paragraph in my mind should really quickly summarize your findings and explain why they're important. Were your hypotheses/predictions met or not?

Our response: We agree and have moved and rewrite this information from the end of the discussion until the start of the discussion (lines 363-370).

Appendix C

Thank you for the opportunity to review this revised manuscript. The authors report important information regarding the first age at reproduction in wild wolves of Scandinavia from the period of early wolf recolonization through increasing population levels. The rigor and resolution of the data in this manuscript are usually quite difficult to obtain but through careful, long term monitoring and the application of cutting-edge technologies and modeling, the researchers have gleaned remarkable results.

The findings have significant implications to conservation and management of wild wolves in terms of expected population trajectories (viability analyses, etc.) and also to broader issues such as estimating dog domestication (inter-generational reproduction estimates).

This research has also raised interesting questions that should be addressed in the future in other study areas to flesh out the broader applicability of these results regarding the influence of inbreeding, territorial establishment, and mate availability on age at first reproduction relative to maturation and nutrition. In addition to containing critical information related to wolf research and conservation / management, this manuscript is also very broadly relevant to other species in other ecological situations.

The manuscript is well-written, well-reasoned and the methodological approaches are sound. The manuscript is appropriate in terms of content / scope for the journal.

The authors have appropriately addressed all my suggestions for revisions.

The only difficulty I had was when trying to access the data – [The dataset and code associated with this study are available from the Dryad Digital Repository: doi:10.5061/dryad.3n5tb2rgg](https://doi.org/10.5061/dryad.3n5tb2rgg).

The authors have appropriately addressed all my suggestions for revisions. I found 6 other items from the lead author on that site – but this particular doi link did not work and I could not find the data and code associated with this project.

Appendix D

Dear Prof. Kruuk,

Please see below each comment and our responses (in bold). Line numbers refer to those in the newly revised manuscript with marked changes.

With best regards,

Camilla Wikenros

Associate Editor

Comments to Author:

I agree with the reviewer that this will be a nice contribution, the authors have now addressed all of the comments thoroughly, and have included helpful context of life history theory and wolf biology for management.

Our response: Thank you very much.

Reviewer(s)' Comments to Author:

Referee: 1

Comments to the Author(s).

Please see the attached file.

Thank you for the opportunity to review this revised manuscript. The authors report important information regarding the first age at reproduction in wild wolves of Scandinavia from the period of early wolf recolonization through increasing population levels. The rigor and resolution of the data in this manuscript are usually quite difficult to obtain but through careful, long term monitoring and the application of cutting-edge technologies and modeling, the researchers have gleaned remarkable results.

The findings have significant implications to conservation and management of wild wolves in terms of expected population trajectories (viability analyses, etc.) and also to broader issues such as estimating dog domestication (inter-generational reproduction estimates).

This research has also raised interesting questions that should be addressed in the future in other study areas to flesh out the broader applicability of these results regarding the influence of inbreeding, territorial establishment, and mate availability on age at first reproduction relative to maturation and nutrition. In addition to containing critical information related to wolf research and conservation /management, this manuscript is also very broadly relevant to other species in other ecological situations.

The manuscript is well-written, well-reasoned and the methodological approaches are sound. The manuscript is appropriate in terms of content / scope for the journal.

The authors have appropriately addressed all my suggestions for revisions. The only difficulty I had was when trying to access the data – The dataset and code associated with this study are available from the Dryad Digital Repository: doi:10.5061/dryad.3n5tb2rgg.

The authors have appropriately addressed all my suggestions for revisions. I found 6 other items from the lead author on that site – but this particular doi link did not work and I could not find the data and code associated with this project.

Our response: Thank you very much. We have checked the Dryad reference to our archived data and made it public. We have received an e-mail saying that it will be curated shortly.

Additional edits:

Lines 4, 17-18: we have added a present address for Morgane Gicquel.

Line 8: we have corrected the address due to new postal number from 2021-03-01.

Lines 40: we have clarified the sentence.

Lines 91, 93, 187, 227, 245, 290, 296-297, 369, 452, 454: we have corrected the number format.

Line 149: we have replaced area with system.

Lines 210, 272, 291, 307-308, 313, 319, 326: we have corrected the reference to the electronic supplementary material.

Lines 309-311, 365, 417-419, 423 435: we have corrected the sentence.

We were asked to reduce the paper with 700 words as it exceeded the page limit of 10 printed pages and we have shortened or removed text on lines 151-159, 163-164, 175-180, 188-194, 197-198, 212-217, 243-245, 273, 340-342, 345-352, 369-370, 390, 401-404, 406-409, 426-429, 440, 443-449, 451, 454-455, 457-465, 467-473, 503-504.

Lines 180-181: we have added a reference for further description of the study system.

Line 223: corrected to individuals.

Line 331: we have replaced semi with spatially.

Line 379: we have replaced on with to.

Line 393: we have replaced when with if.

Lines 496-498: we have added Data accessibility.

Table 1: we have corrected some numbers and colors.

Lines 749-768: we have added Ethics statement, Competing interests statement and Authors' contributions statement.

We have removed 7 references.